



# 1 Greenhouse gas fluxes in mangrove forest soil in the Amazon estuary

Saúl Edgardo Martínez Castellón[1], José Henrique Cattanio[1*], José Francisco Berrêdo[1;3],
Marcelo Rollnic[2], Maria de Lourdes Ruivo[1;3], Carlos Noriega[2].
[1] Graduate Program in Environmental Sciences. Federal University of Pará, Belém,
Brazil
[2] Marine Environmental Monitoring Research Laboratory. Federal University of Pará,
Belém, Brazil.
[3] Department of Earth Sciences and Ecology. Paraense Emílio Goeldi Museum, Belém,
Brazil
[*] Corresponding author: cattanio@ufpa.br (J.H. Cattanio)
Abstract: Tropical mangrove forests are important carbon sinks, the soil being the main
reservoir of this chemical element. Understanding the variability and the key factors that
control fluxes is critical to account for greenhouse gas (GHG) emissions, especially in a
scenario of global climate change. The current study is the first to quantify methane
($CH_4$) and carbon dioxide ($CO_2$) emissions using a dynamic chamber in Amazon natural
mangrove soils. Sampling points were selected in a contrasting topographic gradient,
the highest point being where flooding occurs only at high tides during the solstice and
on the high tides of the rainy season of the new and full moons. The results showed that
mangrove soils are sources of greenhouse gases, and $CO_2$ fluxes were not different
between seasons, and only in the dry period were they greater in the high topography.
Only in the low topography, the $CH_4$ fluxes were higher in the rainy season. However,
in the dry period, the low topography soil produced more $CH_4$. Soil organic matter,
carbon and nitrogen ratio (C/N), and redox potential influenced the annual and seasonal
variation of $CO_2$ emissions; however, they did not influence $CH_4$ flux. To account for
global GHG emissions, in the Amazonian estuary mangrove soil produced 35.4 Mg
$CO_{2-eq}$ ha$^{-1}$ yr$^{-1}$.

## 1 Introduction

The Amazon coastal areas in the State of Pará (Brazil) cover 2,176.8 km$^2$ where
mangroves develop under the macro-tide regime in the (Souza Filho, 2005),
representing approximately 85% of the entire area of Brazilian mangroves (Herz, 1991).
These mangrove areas are estimated to be the main contributors to greenhouse gas
emissions in marine ecosystems (Allen et al., 2011; Chen et al., 2012). However,
mangrove forests are highly productive due to a high nutrient turnover rate (Robertson
et al., 1992) and have mechanisms that maximize carbon gain and minimize water loss
through plant transpiration (Alongi and Mukhopadhyay, 2015). A study conducted in 25



mangrove forests (between 30° latitude and 73° longitude) revealed that these forests
are the richest in carbon storage in the tropics, containing on average 1023 Mg C ha$^{-1}$ of
which 49 to 98% is present in the soil (Donato et al., 2011). In addition, phenolic
compounds inhibit microbial activity and help keep organic carbon intact, thus
accumulating organic matter in mangrove forest soils (Friesen et al., 2018).
The production of greenhouse gases from soils is mainly attributable to biogeochemical
processes. Microbial activities and gas production are related to soil properties,
including total carbon and total nitrogen concentrations, moisture, porosity, salinity, and
redox potential (Bouillon et al., 2008; Chen et al., 2012). Due to the dynamics of tidal
movements, mangrove soils may become saturated and present a reduced oxygen
availability or total aeration caused by the ebb tide. Studies attribute soil carbon flux
responses to moisture perturbations because of seasonality and flooding events
(Banerjee et al., 2016), with fluxes being dependent on tidal extremes (high tide and low
tide), and flood duration (Chowdhury et al., 2018).
The estimated $CO_2$ production to the atmosphere, in tropical estuarine areas, is 16.2
TgCy$^{-1}$ (Alongi, 2009). However, the most recent estimate between latitude 0° to 23.5°
S reveals an emission of 2.3 g $CO_2$ m$^{-2}$ d$^{-1}$ (Rosentreter et al., 2018a). In situ $CO_2$
production is related to the water input of terrestrial, riparian, and groundwater brought
by rainfall (Rosentreter et al., 2018c).
Due to this periodic tidal influence, the mangrove ecosystem is regularly flooded,
leaving the soil anoxic and reduced, favoring methanogenesis (Dutta et al., 2013). Thus,
estuaries are considered hot spots for $CH_4$ production and emission (Bastviken et al.,
2011; Borges et al., 2015). The organic material decomposition by methanogenic
bacteria in anoxic environments, such as sediments, inner suspended particles,
zooplankton gut (Reeburgh, 2007; Valentine, 2011), and the reduction of sulfate in
anoxic marine sediments (Purvaja et al., 2004), also results in $CH_4$ formation. On the
other hand, an ecosystem with salinity levels greater than 18 ppt may show an absence
of $CH_4$ emissions (Poffenbarger et al., 2011). Currently the uncertainties in emitted $CH_4$
values in vegetated coastal wetlands are approximately 30% (EPA, 2017). The total
emission of 0.010 Tg $CH_4$ y$^{-1}$ or 0.64 g CH4 m$^{-2}$ d$^{-1}$ was estimated between 0 and 5°
latitude (Rosentreter et al., 2018a).
The objective of this study is to investigate the spatial and seasonal variation in the
monthly fluxes of $CO_2$ and $CH_4$ from the soil in a non-anthropized mangrove area in the





Mojuim River Estuary, belonging to the Amazon biome. The environmental factors and
physicochemical analysis of the soil were investigated from 2017 to 2018 to understand
the gas fluxes.
**2    Material and Methods**
2.1    Study site
This study was conducted in the Amazonian coastal zone, Macaca Island, located in the
Mojuim River estuary, at the Mocapajuba Marine Extractive Reserve, municipality of
São Caetano de Odivelas (Figure 1), state of Pará (Brazil). Macaca island has an area of
1,322 ha with exclusively untouched mangrove forests, which belongs to a coastal strip
of 2177 km$^2$ in the state of Pará (Souza Filho, 2005). The climate is Am type according
to the Köppen classification (Peel et al., 2007). The climatological data were obtained
from the Meteorological Database for Teaching and Research of the National Institute
of Meteorology (INMET). The area has a rainy season from January to June (2,296 mm
of precipitation) and a dry season from July to December (687 mm). March and April
are the rainiest months with 505 and 453 mm of precipitation, while October and
November are the driest (53 and 61 mm, respectively). The minimum temperatures
occur in the rainy period (26 °C) and the maximum in the dry period (29 °C). The
Mojuim estuary has a macrotidal regime, with an average height of 4.9 m during spring
tide and 3.2 m during low tide (Rollnic et al., 2018). During the wet season the Mojuim
River has a flow velocity of 1.8 m s$^{-1}$ at the ebb tide and 1.3 m s$^{-1}$ at the flood tide.
During the dry season, the maximum currents are 1.9 m s$^{-1}$ at the flood and 1.67 m s$^{-1}$ at
the ebb tide (Rocha, 2015) The annual mean salinity is 26.95 ± 0.98 PSU (Valentim et
al., 2018).



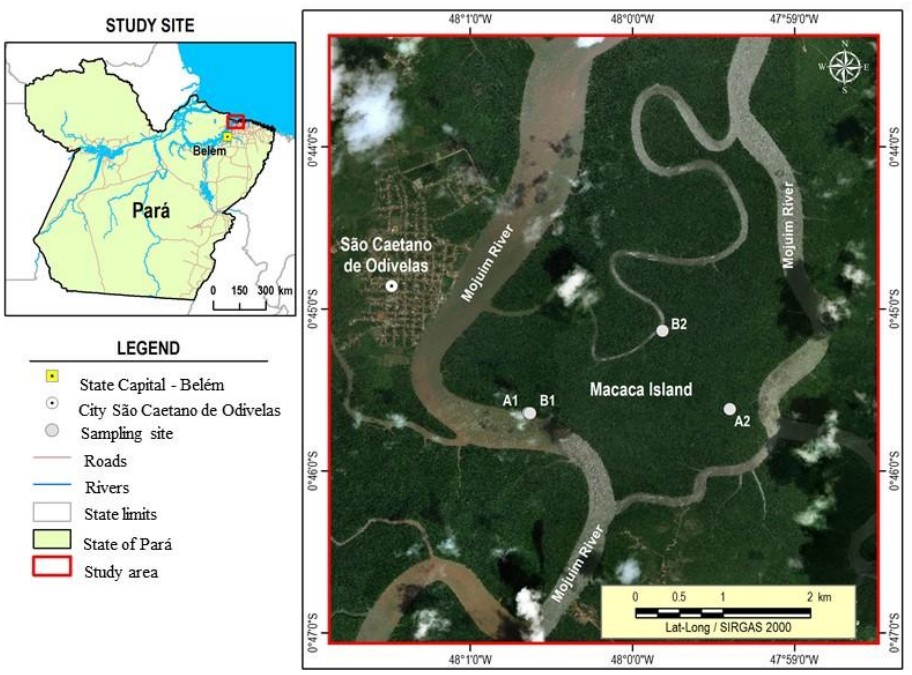

Figure 1. Macaca Island located in the mangrove coast of Northern Brazil, Municipality of São Caetano de Odivelas (state of Pará), with the sampling points at low (B1 and B2) and high topographies (A1 and A2). Image Source: © Google Earth

The Mojuim River region is geomorphologically formed by partially submerged river basins consequent of an increase in the relative sea level during the Holocene (Prost et al., 2001) associated with the formation of mangroves, dunes, and beaches (El-Robrini et al., 2006). This river forms the entire watershed of the municipality of São Caetano de Odivelas and borders the municipality of São João da Ponta (Figure 1). Before reaching the estuary, the Mojuim River crosses an area of a dryland forest highly fragmented by family farming, forming remnants of secondary forest (< 5.0 ha) with various ages (Fernandes and Pimentel, 2019). The population economically exploited the estuary, primarily by artisanal fishing, crab (*Ucides cordatus* L.) extraction, and oyster farms.

Four sampling sites were selected in the Macaca Island: two where flooding occurs every day (B1 and B2; Figure 1), called low topography, and two where flooding occurs only at high tides during the solstice and on the high tides of the rainy season of the new and full moons (A1 and A2; Figure 1), called high topography. Once a month, the gas



flux for each chamber was measured during periods of waning or crescent moon, as
these are the times when the soil in the low topography is more exposed. The flora of
the mangrove area on the Macaca Island is little anthropized and comprises the genera
*Rhizophora*, *Avicenia*, *Laguncularia*, and *Acrostichum* (Ferreira, 2017; França et al.,
2016). The estuarine plains are influenced by a macro tide dynamics and can be
physiographically divided into four sectors (França et al., 2016). The Macaca Island is
classified as being from the fourth sector, which consists of woods of adult trees of the
genus *Ryzophora* with an average height of 10 to 25 m, located at an elevation of 0 to 5
m, with silt-clay soil (França et al., 2016).
2.2    Vegetation structure and biomass
The floristic survey was conducted at the same sites as the gas flow study, using circular
plots of 1,256.6 m$^2$ (Kauffman et al., 2013), divided into four subplots of 314.15 m$^2$,
which is the equivalent to 0.38 ha (Figure 1). All trees with DBH (diameter at breast
height) greater than 0.05m had their diameter above the aerial roots, the diameter of the
stem, and total height recorded. The allometric equation to calculate tree biomass
(AGB) was: $AGB = 0.168 \times \rho \times (DBH) \times 2.471$, where ρ represents wood density,
using 0.87 g cm$^{-3}$ for *R. mangle* and 0.72 g cm$^{-3}$ for *A. germinans* (Howard et al.,
2014b).
2.3    Soil sampling and environmental characterization
In July 2017 and January 2018, four soil samples were collected with an auger at a
depth of 0.10 m in all the studied sites (Figure 1). Before the soil samples were
removed, pH and redox potential (Eh; mV) were measured with a Metrohm 744
equipment by inserting the platinum probe directly into the soil at a depth of 0.10 m
(Bauza et al., 2002). The soil samples were properly stored and taken to the Chemical
Analysis Laboratory of the *Museu Paraense Emílio Goeldi*. Salinity (Sal; ppt) was
measured with PCE-0100, and soil moisture (Sm; %) by the residual gravimetric
method (EMBRAPA, 1997).
Organic Matter (OM; g kg$^{-1}$), Total Carbon (TC; g kg$^{-1}$) and Total Nitrogen (TN; g kg$^{-1}$)
were calculated by volumetry (oxidoreduction) using the Walkley-Black method
(Kalembasa and Jenkinson, 1973). Microbial carbon (Cmic; mg kg$^{-1}$) and microbial
nitrogen (Nmic; mg kg$^{-1}$) were determined through the 2,0 min of Irradiation-extraction
method of soil by microwave technique (Islam and Weil, 1998). Microwave heated soil





extraction proved to be a simple, fast, accurate, reliable and safe method to measure soil
microbial biomass (Araujo, 2010; Ferreira et al., 1999; Monz et al., 1991). The Cmic
was determined by dichromate oxidation (Kalembasa and Jenkinson, 1973; Vance et al.,
1987). The Nmic was analyzed following the method described by Brookes et al.
(1985), changing fumigation to irradiation, which uses the difference between the
amount of TN in irradiated and non-irradiated soil. We used the flux conversion factor
of 0.33 (Sparling and West, 1988) and 0.54 (Almeida et al., 2019; Brookes et al., 1985),
for carbon and nitrogen, respectively. Particle size analysis was performed separately on
four soil samples collected at each flux site, in the two seasons, according to
EMBRAPA (1997). .
At each flow measurement, environmental variables such as air temperature (Tair, °C),
relative humidity (RH, %), wind speed (Ws, m s$^{-1}$) were quantified with a portable
thermo-hygrometer (model AK821) at the height of 2.0 m above the soil surface. Soil
temperature (Ts, °C) was measured with a portable digital thermometer (model TP101)
sequentially after each flow measurement. Daily precipitation was obtained from an
automatic precipitation station installed at a pier on the banks of the Mojuim River in
São Caetano das Odivelas (coordinates: 0°44'18.48 "S; 48°00'47.94 "W).
2.4    Fluxes Measurements
In each plot, eight Polyvinyl Chloride rings with 0.20 m diameter and 0.12 m height
were randomly installed within a circumference with a diameter of 20 m. The rings had
an area of 0.028 m$^{-2}$ (volume of 3.47 L) and were fixed 0.05 m into the ground. The
height of the ring above ground was measured at four equidistant points with a ruler at
each flow measurement. To avoid the influence of mangrove roots on the gas fluxes, the
rings were placed in locations without any seedlings or aboveground mangrove roots.
$CO_2$ and $CH_4$ fluxes (g $CO_2$ or $CH_4$ m$^{-2}$ d$^{-1}$) were measured using the dynamic chamber
methodology (Norman et al., 1997; Verchot et al., 2000), sequentially connected to a
Los Gatos Research portable gas analyzer (Mahesh et al., 2015). The device was
calibrated monthly with high quality standard gas. The rings were sequentially closed
for three minutes with a PVC cap, which enabled the connection to the analyzer via two
12.0 m polyethylene hoses. The gas concentration was measured (ppm) every two
seconds and automatically stored by the analyzer. $CO_2$ and $CH_4$ fluxes were calculated
from the linear regression of increasing/decreasing $CO_2$ and $CH_4$ concentrations within
the chamber, usually between one and three minutes after the ring cover was placed





(Frankignoulle, 1988; McEwing et al., 2015). Analyzing the literature, we found that the
flux is considered zero when the linear regression reaches an $R^2 < 0.30$ (Sundqvist et al.,
2014). However, in our analyses, the vast majority of regressions reached an $R^2 > 0.70$,
and the regressions were weak in only 6% of the data.
2.5    Statistical analyses
The normality of the data of $FCH_4$ and $FCO_2$ and soil physicochemical parameters was
determined by the Shapiro-Wilks method. The student's t-test was used to test the
differences ($p < 0.05$) in the emissions between the different sites and seasonal periods.
An ANOVA and Tukey's test ($p < 0.05$) were used when the distributions were normal.
For non-parametric data the Kruskal-Wallis test was used ($p < 0.05$). Pearson
correlation coefficients were calculated to determine the relationships between soil
properties and gas fluxes. Statistical analyses were performed with and free statistical
software Infostat 2015®.
**3    Results**
3.1    Precipitation
There was a marked seasonality during the study period (Figure 2), with 2,155.0 mm of
precipitation during the rainy period and 1,016.5 mm during the dry period. However,
the rainfall distribution was different from the climatological average (Figure 2). The
rainy season had 553.2 mm less precipitation, and the dry season had 589.1 mm more
than the climatological normal. Thus, in the period studied, the dry season was rainier,
and the rainy season was drier than the climatological normal.



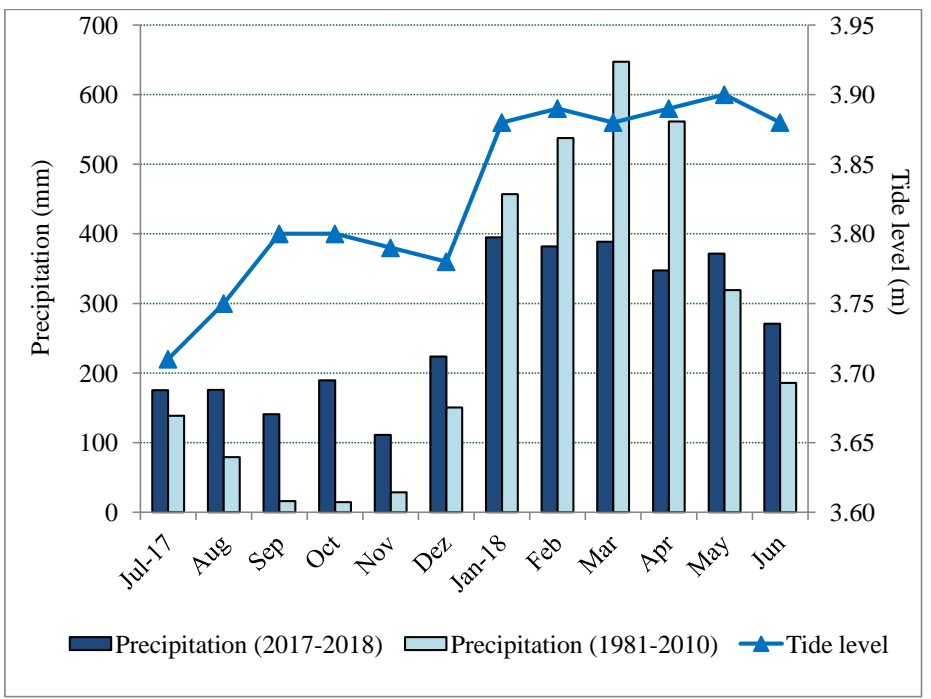


Figure 2. Monthly climatological normal in the municipality of Soure (1981-2010, mm), monthly precipitation (mm), and maximum tide height (m) for the from 2017 to 2018, in the municipality of São Caetano de Odivelas (PA).

3.2    Carbon Dioxide and methane efflux

The $CO_2$ and $CH_4$ fluxes in the mangrove soil were not normally distributed, so the statistical analysis was performed using a non-parametric method. $CO_2$ fluxes only differed among topographies in January (H = 3.915; p = 0.048), July (H = 9.091; p = 0.003), and November (H = 11.294; p < 0.000) (Table 1), with generally higher fluxes at the high topography than at the low topography. $CH_4$ fluxes were statistically different between topographies only in November (H = 9.276; p = 0.002) and December (H = 4.945; p = 0.005), with higher fluxes at the low topography (Table 1)

Table 1. Monthly and seasonal (dry and rainy seasons) fluxes of $CO_2$ and $CH_4$ (g $CO_2$ or $CH_4$ m$^{-2}$ d$^{-1}$) at the high and low topographies. Numbers represent the mean (standard error). Lower case letters compare topographies in the same month. Upper case letters compare stations at each topography. Different boldface letters have statistically significant variation (Kruskal Wallis, p < 0.05).



| | $CO_2$ flux (g m$^{-2}$ d$^{-1}$) | | $CH_4$ flux (g m$^{-2}$ d$^{-1}$) | |
|---|---|---|---|---|
| | High topography | Low topography | High topography | Low topography |
| July/2017 | **10.166(1.555)[a]** | **4.036(1.027)[b]** | 0.0724(0.0518)[a] | 0.2129(0.2087)[a] |
| August | 8.513(2.672)[a] | 12.462(3.400)[a] | 0.0033(0.0016)[a] | 0.1270(0.1185)[a] |
| September | 11.506(2.515)[a] | 6.020(1.207)[a] | 0.0014(0.0008)[a] | 0.1738(0.1608)[a] |
| October | 4.147(0.653)[a] | 3.993(0.731)[a] | 0.0000(0.0000)[a] | -0.0004(0.0056)[a] |
| November | **7.648(1.064)[a]** | **0.007(0.002)[b]** | **-0.0004(0.0001)[b]** | **0.1395(0.0708)[a]** |
| December | 5.302(1.176)[a] | 7.622(2.505)[a] | **0.0009(0.0009)[b]** | **0.1210(0.0575)[a]** |
| Dry period | **7.902(0.803)[aA]** | **6.202(0.895)[bA]** | **0.0141(0.010)[bB]** | **0.1280(0.053)[aA]** |
| January/2018 | **6.697(1.717)[a]** | **2.995(0.493)[b]** | 0.0007(0.0004)[a] | 0.0294(0.0183)[a] |
| February | 9.053(2.650)[a] | 6.384(1.428)[a] | 0.0049(0.0022)[a] | 0.8743(0.7024)[a] |
| March | 5.225(1.135)[a] | 5.970(1.534)[a] | 0.0077(0.0056)[a] | 0.3736(0.2197)[a] |
| April | 14.077(4.695)[a] | 4.785(0.711)[a] | 0.1968(0.1304)[a] | 0.0372(0.2841)[a] |
| May | 3.299(0.587)[a] | 3.565(0.472)[a] | 0.0014(0.0019)[a] | 0.0218(0.5648)[a] |
| June | 8.796(2.053)[a] | 4.704(1.183)[a] | 0.0226(0.0191)[a] | 0.6739(0.6665)[a] |
| Rainy period | 7.858(1.058)[aA] | 4.734(0.440)[aA] | **0.0390(0.023)[aA]** | 0.3350(0.194)[aA] |

At the high topography, $CO_2$ fluxes were significantly higher in July compared to
August and December, March, October, and May, not differing from the other months
of the year (H = 24.510; p = 0.011). $CH_4$ fluxes at the high topography were
significantly (H = 40.073; p < 0.001) higher in April and July compared to the other
months studied, and in November there was consumption of $CH_4$ from the atmosphere
(Table 1). At the low topography, $CO_2$ fluxes were statistically (H = 19.912; p = 0.046)
higher in September and February compared to January and November, not differing
from the other months. $CH_4$ fluxes at the low topography did not show a significant
variation between months (H = 10.114; p = 0.407).
Although seasonal $CO_2$ fluxes were higher at the high topography than at the low
topography (Table 1), they were only statistically different in the dry season (H = 7.378;
p = 0.006). In contrast, seasonal $CH_4$ fluxes were higher at the low topography (Table 1)
but were only statistically different in the dry season (H = 8.229; p < 0.001). With this
the mean annual fluxes of $CO_2$ and $CH_4$ were 6.659 ± 0.419 g $CO_2$ m$^{-2}$ d$^{-1}$ (mean ±
standard error) and 0.132 ± 0.053 g $CH_4$ m-2 d-1, respectively.





### 3.3 Environmental characterization

Silt concentration was higher at the low topography (LSD: 14.763; p= 0.007) and clay concentration was higher at the high topography sites (LSD: 12.463; p= 0.005), in both stations studied (Table 2). Soil particle size analysis did not vary statistically (p > 0.05) between the two stations (Table 2). Soil moisture did not vary significantly (p > 0.05) between topographies at each station, or between seasonal periods at the same topography (Table 2). The variable pH varied statistically only at the low topography when the two stations were compared (LSD: 5.950; p= 0.006), being more acidic in the dry period (Table 2). On average pH was significantly (LSD: 0.559; p= 0.008) higher in the dry season (Table 2). No variation in Eh was identified between topographies and seasons (Table 2), although it was higher in the dry season than in the rainy season. However, Sal values were higher (LSD: 3.444; p = 0.010) at the high topography than at the low topography in the dry season (Table 2). In addition, Sal was significantly higher in the dry season in both the high (LSD: 2.916; p < 0.001) and low (LSD: 3.003; p < 0.001) topographies (Table 2).





Table 2. Concentration analysis of Sand, Silt, Clay, Moisture, pH, Redox Potential (Eh) and salinity (Sal; ppt) in mangrove soil in the high and
low topographies, and in the rainy and dry seasons, at Macaca island, São Caetano das Odivelas. Numbers represent the mean (standard error of
the mean). Lower case letters compare topographies in each seasonal period, and upper-case letters compare the same topography between
seasonal periods. Different letters indicate statistical variation (LSD, $p < 0.05$).

| Season | Topography | Sand (%) | Silt (%) | Clay (%) | Moisture (%) | pH | Eh (mV) | Sal (ppt) |
|---|---|---|---|---|---|---|---|---|
| Dry | High | 12.1(1.4)[aA] | 41.8(3.3)[bA] | 46.1(2.6)[aA] | 73.1(6.6)[aA] | 5.5(0.2)[aA] | 190.25(45.53)[aA] | 35.25(1.11)[aA] |
| | Low | 9.7(2.5)[aA] | 63.6(6.1)[aA] | 26.6(5.2)[bA] | 86.9(3.4)[aA] | 5.3(0.3)[aA] | 106.38(53.76)[aA] | 30.13(1.16)[bA] |
| | Mean | 10.9(1.4)[A] | 52.7(4.4)[A] | 36.4(3.8)[A] | 80.0(4.0)[A] | 5.4(0.2)[A] | 148.31(35.71)[A] | 32.69(1.02)[A] |
| Rainy | High | 12.1(1.4)[aA] | 41.8(3.3)[bA] | 46.1(2.6)[aA] | 88.9(3.5)[aA] | 4.9(0.4)[aA] | 92.50(56.20)[aA] | 7.50(0.78)[aB] |
| | Low | 9.7(2.5)[aA] | 63.6(6.1)[aA] | 26.6(5.2)[bA] | 88.6(3.7)[aA] | 4.4(0.1)[aB] | 36.25(49.97)[aA] | 8.13(0.79)[aB] |
| | Mean | 10.9(1.4)[A] | 52.7(4.4)[A] | 36.4(3.8)[A] | 88.7(2.5)[A] | 4.6(0.2)[B] | 64.38(37.04)[A] | 7.81(0.54)[B] |






The Cmic did not differ between topographies in the two seasons (Table 3); however
CT was significantly higher in the low topography in the dry season (LSD: 5.589; p <
0.000) and in the rainy season (LSD: 5.777; p = 0.024). In addition, Cmic was higher in
the dry season in both the high (LSD: 11.325; p < 0.010) and low (LSD: 9.345; p <
0.000) topographies (Table 3). Nmic did not vary between topographies seasonally.
However, Nmic in the high (LSD: 9.059; p = 0.013) and low topographies (LSD: 4.447;
p = 0.001) was higher during the dry season (Table 3). The C/N ratio (Table 3) was
higher in the low topography in both the dry (LSD: 3.142; p < 0.000) and rainy seasons
(LSD: 3.675; p = 0.033), when compared to the high topography. However, only in the
low topography was the C/N ratio higher (LSD: 1.863; p < 0.000) in the dry season
compared to the rainy season (Table 3). Soil MO was higher at the low topography in
the rainy (LSD: 9.950; p = 0.024) and in the dry seasons (LSD: 9.630; p < 0.000).
However, only in the lowland topography was the MO concentration higher in the dry
season than in the rainy season (Table 3).





Biogeosciences Discussions — Open Access — EGU

Table 3. Seasonal and topographic variation in microbial Carbon (Cmic; mg kg⁻¹), microbial Nitrogen (Nmic, mg kg⁻¹), Total Carbon (TC; g kg⁻
¹), Total Nitrogen (NT; g kg⁻¹), Carbon/Nitrogen ratio (C/N) and Soil Organic Matter (OM; g kg⁻¹). Numbers represent the mean (standard error).
Lower case letters compare topography at each station, and upper-case letters compare topography among stations.

| Season | Topography | $C_{mic}$ mg kg⁻¹ | $N_{mic}$ mg kg⁻¹ | $C_T$ g kg⁻¹ | $N_T$ g kg⁻¹ | C/N | OM g kg⁻¹ |
|---|---|---|---|---|---|---|---|
| Dry | High | 22.12(5.22)[aA] | 12.76(4.20)[aA] | 14.12(2.23)[bA] | 1.43(0.06)[aA] | 9.60(1.20)[bA] | 24.35(3.84)[bA] |
| | Low | 26.34(4.23)[aA] | 10.34(2.05)[aA] | 26.44(1.35)[aA] | 1.56(0.04)[aA] | 16.98(0.84)[aA] | 45.59(2.32)[aA] |
| | Mean | 24.23(3.29)[A] | 11.55(2.28)[A] | 20.28 (2.03)[A] | 1.49(0.04)[A] | 13.29(1.19)[A] | 34.97(3.50)[A] |
| Rainy | High | 7.40(0.79)[aB] | 0.75(0.41)[aB] | 11.46(2.48)[bA] | 1.32(0.04)[aA] | 8.42(1.70)[bA] | 19.75(4.27)[bA] |
| | Low | 5.95(1.06)[aB] | 1.23(0.28)[aB] | 18.27(1.06)[aB] | 1.46(0.06)[aA] | 12.47(0.22)[aB] | 31.51(1.83)[aB] |
| | Mean | 6.68(0.67)[B] | 0.99(0.25)[B] | 14.86 (1.57)[B] | 1.39(0.04)[A] | 10.44(0.98)[A] | 25.63(2.71)[B] |




Tar was significantly higher (LSD = 0.72, p = 0.01) at the high topography (31.24 ±
0.26 °C) than at the low topography (30.30 ± 0.25 °C) only in the rainy season (Figure
3a). No significant variation in Ts was found between the topographies in both seasons
(Figure 3b). The RH was significantly higher (LSD = 2.55, p = 0.01) at the high
topography (70.54 ± 0.97%) than at the low topography (66.85 ± 0.87%) only in the
rainy season (Figure 3c). At this same station, Vv (Figure 3d) was significantly higher
(LSD = 0.15, p < 0.00) at the low topography (0.54 ± 0.06 m s$^{-1}$) than at the high
topography (0.24 ± 0.04 m s$^{-1}$).

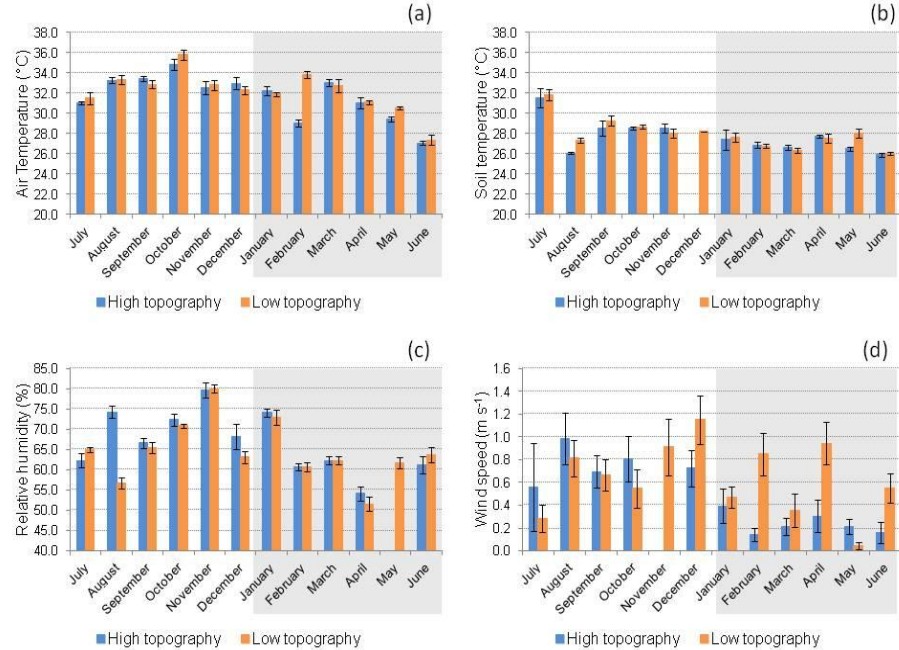


Figure 3. a) Air temperature (°C), b) soil temperature (°C), c) relative humidity (%) and
d) wind speed (m s$^{-1}$) at high and low topographies, from July 2017 to June 2018 in a
mangrove area in the Mojuim River estuary. Bars highlighted in grey correspond to the
rainy season (n = 16). The bars represent the standard error.
3.4   Vegetation structure and biomass
Only the species *R. mangle* and *A. germinans* were found in the floristic survey carried
out. The DBH did not vary significantly between the topographies for either species
(Table 4). However, *R. mangle* had a higher DBH than *A. germinaris* at both high



(LSD: 139.304; p = 0.037) and low topographies (LSD: 131.307; p = 0.001). The basal
area (BA) and AGB variables did not show significant variation (Table 4). A total
aboveground biomass of 322.1 ± 49.6 Mg ha$^{-1}$ was estimated.





Table 4: Sum of Diameter at Breast Height (DBH), Basal Area (BA) and Above Ground Biomass (AGB) at high and low topography in the
mangrove forest of the Mojuim River estuary. Numbers represent the mean (standard error of the mean). Lower case letters compare topographic
height for each species, and upper-case letters compare species at each topographic height, using Tuckey's test (p < 0.05).

| Specie | Topography | N ha$^{-1}$ | DBH (cm) | BA (m$^2$ ha$^{-1}$) | AGB (Mg ha$^{-1}$) |
|---|---|---|---|---|---|
| *Rhizophora mangle* | High | 302.4(20.5) | 238.8(24.9)[aA] | 17.3(2.0)[aA] | 219.3(25.7)[aA] |
| | Low | 310.4(37.6) | 283.5(45.0)[aA] | 24.2(4.3)[aA] | 338.7(62.9)[aA] |
| *Avicennia germinans* | High | 47.7(20.5) | 86.8(51.2)[aB] | 13.8(9.2)[aA] | 135.3(94.7)[aA] |
| | Low | 15.9(9.2) | 46.1(29.3)[aB] | 11.8(8.8)[aA] | 136.0(108.3)[aA] |
| Total | High | 350.2(18.4) | 325.6(33.6)[a] | 31.1(7.5)[a] | 304.5(99.8)[a] |
| | Low | 346.2(41.0) | 296.0(23.7)[a] | 30.0(4.1)[a] | 330.8(60.4)[a] |

The equations for biomass estimates (AGB) were: *R. mangle* = 0.1282*DBH$^{2.6}$, *A. germinans* = 0.14*DBH$^{2.4}$, Total = 0.168*ρ*DBH$^{2.47}$, where ρ$_{R.\ mangle}$ = 0.87; ρ$_{A.\ germinans}$ =
0.72 (Howard et al., 2014a).





## 4   Discussion

### 4.1   Carbon dioxide e methane flux measurements

It is important to consider that the year under study was rainier in the dry season and less rainy in the wet season (Figure 2). Perhaps this variation is already related to the effects of global climate changes. Under these conditions, the $CO_2$ flux from the mangrove soil ranged from -5.06 to 68.96 g $CO_2$ m$^{-2}$ d$^{-1}$ (mean 6.66 g $CO_2$ m$^{-2}$ d$^{-1}$), while the $CH_4$ flux ranged from -5.07 to 11.08 g $CH_4$ m$^{-2}$ d$^{-1}$ (mean 0.13 g $CH_4$ m$^{-2}$ d$^{-1}$), resulting in a total carbon rate of 7.04 g $CO_2$ m$^{-2}$ d$^{-1}$ or 25.70 Mg $CO_2$ ha$^{-1}$ y$^{-1}$ (negative values represent atmospheric consumption of the gas) (Figure 4). The negative $CO_2$ flux appears to be a consequence of the increased $CO_2$ solubility in tidal waters, or of the increased sulfate reduction (Borges et al., 2018; Chowdhury et al., 2018; Nóbrega et al., 2016). The soil carbon flux in mangrove area in the Amazon region was within the range of findings for other tropical mangrove areas (2.57 to 11.00 g $CO_2$ m$^{-2}$ d$^{-1}$; Shiau and Chiu, 2020). However, the mean flux of 6.2 mmol $CO_2$ m$^{-2}$ h$^{-1}$ recorded in this Amazonian mangrove was much higher than the mean efflux of 2.9 mmol $CO_2$ m$^{-2}$ h$^{-1}$ recorded in 75 mangroves during low tide periods (Alongi, 2009). We found a mean monthly flux of 327.9 ± 78.0 mg $CO_2$ m$^{-2}$ h$^{-1}$ and 217.2 ± 51.0 mg $CO_2$ m$^{-2}$ h$^{-1}$, at the high and low topography, respectively.



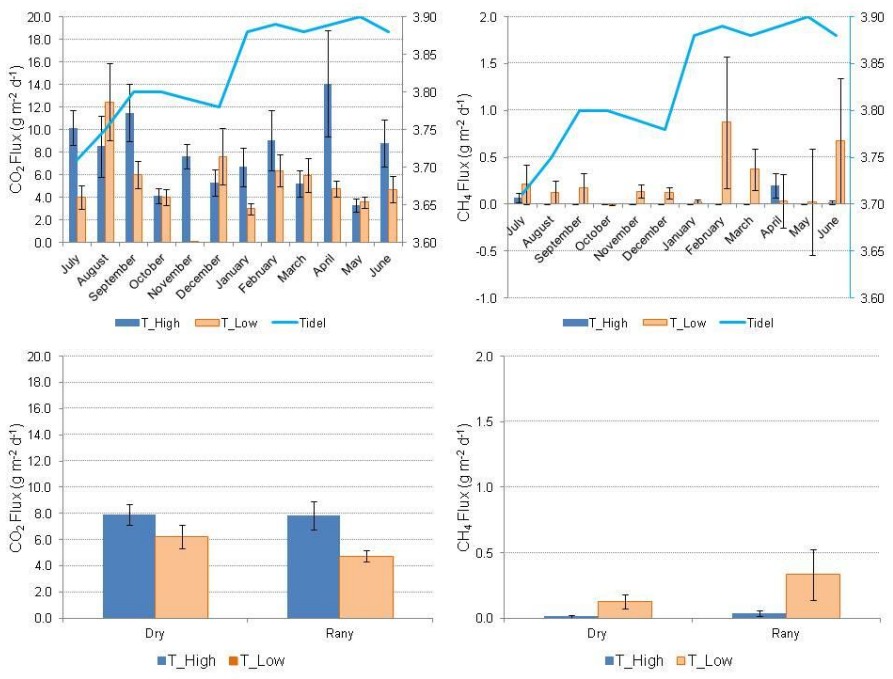

309

Figure 4. $CO_2$ and $CH_4$ fluxes (g $CO_2$ or $CH_4$ m$^{-2}$ d$^{-1}$) monthly (July 2018 to June 2019)

(n = 16) and seasonally (Dry and Rany), at high (T_High) and low (T_Low)

topographies (n = 96), in mangrove forest soil compared to tide level (TideL). The bars

represent the standard error of the mean.

An emission of 0.010 Tg $CH_4$ y$^{-1}$, 0.64 g $CH_4$ m$^{-2}$ d$^{-1}$ (Rosentreter et al., 2018b), or 26.7

mg $CH_4$ m$^{-2}$ h$^{-1}$ is estimated at tropical latitudes (0 and 5°). In our study, the monthly

average in $CH_4$ flux was higher at the low topography (7.3 ± 8.0 mg C m$^{-2}$ h$^{-1}$) than at

the high topography (0.9 ± 0.6 mg C m$^{-2}$ h$^{-1}$) (Figure 4). Therefore, the $CH_4$-C fluxes

from the mangrove soil in the Mojuim River estuary were much lower than expected.

The average emission from the soil of 8.4 mmol $CH_4$ m$^{-2}$ d$^{-1}$ was well below the fluxes

recorded in the Bay of Bengal, with 18.4 mmol $CH_4$ m$^{-2}$ d$^{-1}$ (Biswas et al., 2007). In the

Amazonian mangrove studied the mean annual carbon equivalent efflux was 429.6 mg

$CO_{2\text{-eq}}$ m$^{-2}$ h$^{-1}$. This value is 0.00004% of the erosion losses of 103.5 Tg $CO_{2\text{-eq}}$ ha$^{-1}$ y$^{-1}$

projected for the next century in tropical mangrove forests (Adame et al., 2021).

4.2    Mangrove biomass

Assuming that the amount of carbon stored is 0.42 of the total biomass (Sahu and

Kathiresan, 2019), the mangrove forest biomass of the Mojuim River estuary stores



127.9 and 138.9 Mg C ha[-1] at the high and low topography, respectively. This result is
well below the 507.8 Mg C ha[-1] estimated for Brazilian mangroves (Hamilton and
Friess, 2018), but are near the 103.7 Mg C ha[-1] estimated for a mangrove at Dos Guarás
island (Salum et al., 2020), 108.4 Mg C ha[-1] for the Bragantina region (Gardunho,
2017), and 132.3 Mg Mg C ha[-1] in French Guiana (Fromard et al., 1998). The estimated
primary production for tropical mangrove forests is $218 \pm 72$ Tg C y (Bouillon et al.,
2008). These results show that the mangroves of the Amazon estuary are more
productive than previously known (Bouillon et al., 2008).
4.3    Topography variation
The mangrove areas are periodically flooded, with a larger flood volume during the ebb
tides, especially in the rainy season. The hydrological condition of the soil is determined
by the microtopography and can regulate the respiration of microorganisms (aerobic or
anaerobic), being a decisive factor in controlling the $CO_2$ efflux (Dai et al., 2012;
Davidson et al., 2000; Ehrenfeld, 1995). In the two climatic periods of the year, the high
topography produced more $CO_2$ ($7,869 \pm 1,873$ g $CO_2$ m[-2] d[-1]) than the low topography
($5,212 \pm 1,225$ g $CO_2$ m[-2] d[-1]) (Figure 4). No significant influence on $CO_2$ flux was
observed due to the low variation in high tide level throughout the year (0.19 m) (Figure
4), although it was numerically higher at the high topography. However, tidal height
and the rainy season resulted in a higher $CO_2$ flux (rate high/low =1.7) at the high
topography ($7.858 \pm 0.039$ g $CO_2$ m[-2] d[-1]) than at the low topography ($4.734 \pm 0.335$ g
$CO_2$ m[-2] d[-1]) (Figure 4; Table 1). This result is because the root systems of most flood-
tolerant plants remain active when flooded (Angelov et al., 1996). Still, the high
topography has longer flood-free periods, because this only happens when the tides are
in the form of triangle or when the rains are torrential.
$CO_2$ efflux was higher in the high topography than in the low topography, i.e., 39.8%
lower in the forest soil exposed to the atmosphere for less time, in the rainy season
(when soils are more subject to inundation). Measurements performed on 62 mangrove
forest soils showed an average flux of 2.87 mmol $CO_2$ m[-2] h[-1] when the soil is exposed
to the atmosphere, while 75 results on flooded mangrove forest soils showed an average
emission of 2.06 mmol $CO_2$ m[-2] h[-1] (Alongi, 2007, 2009), i.e., 28.2% less than for the
dry soil. Reflecting the more significant facility gases have for molecular diffusion than
fluids, and the increased surface area for aerobic respiration and chemical oxidation
during air exposure (Chen et al., 2010). Some studies attribute this variation to the



temperature of the soil when exposed to tropical air (Alongi, 2009), increasing the
export of dissolved inorganic carbon (Maher et al., 2018). However, although there was
no significant variation in soil temperature between topographies at each time of year
(Figure 3b), there was a positive correlation (Pearson = 0.15, p = 0.05) between $CO_2$
efflux and soil temperature at the low topography.
In the rainy season, $CO_2$ efflux was correlated with Tar (Pearson = 0.23, p = 0.03), RH
(Pearson = -0.32, p < 0.00) and Ts (Pearson = 0.21, p = 0.04) only at the low
topography. In the dry season $CO_2$ flux was correlated with Ts (Pearson = 0.39, p <
0.00) at low topography. Some studies show that $CH_4$ efflux is a consequence of the
seasonal temperature variation in mangrove forest in temperate/monsoon climate
(Chauhan et al., 2015; Purvaja and Ramesh, 2001; Whalen, 2005). However, in this
study $CH_4$ efflux was correlated with Ta (Pearson = -0.33, p < 0.00) and RH (Pearson =
0.28, p = 0.01) only in the dry season and at the low topography. These results show
that hardly does only one physical parameter interfere with the fluxes, and that they do
not interact similarly in a different topography and seasonality.
A compilation of several studies showed that the total $CH_4$ emissions from soil in
mangrove ecosystem range from 0 to 23.68 mg C m$^{-2}$ h$^{-1}$ (Shiau and Chiu, 2020), and
our study showed a range of -0.01 to 31.88 mg C m$^{-2}$ h$^{-1}$, with a mean of 4.70 ± 5.00 mg
C m$^{-2}$ h$^{-1}$. The monthly $CH_4$ fluxes were generally higher at the low (0.232 ± 0.256)
than at the high (0.026 ± 0.018) topography, especially during the high tide (Figure 4).
Compared to the high topography, only in the dry season was there a significantly
higher production at the low topography (Table 1, Figure 4). The low topography
produced 0.0249 g C m$^{-2}$ h$^{-1}$ more to the atmosphere in the rainy season than in the dry
season (Table 1), and the same seasonal variation was recorded in other studies
(Cameron et al., 2021).
The mangrove soil in the Mojuim River estuary is rich in silt and clay (Table 2), which
reduces sediment porosity and fosters the formation and retention of anoxic conditions
(Dutta et al., 2013). In addition, the lack of oxygen in the flooded mangrove soil
generates microbial processes such as denitrification, sulfate reduction, methanogenesis,
and redox reactions (Alongi and Christoffersen, 1992). Furthermore, plenty of the $CH_4$
produced in wetlands is dissolved in situ in the pore water caused by the high pressure,
which can result in supersaturation in the water, enabling $CH_4$ to be released from the
sediment to the atmosphere by diffusion and by boiling in the water (Neue et al., 1997).





Only the species *R. mangle* and *A. germinans* were found in the floristic survey carried
out, which agrees with other studies in the same region (Menezes et al., 2008). Thus, the
variations found in the flux between the topographies in the Mojuim River estuary are
not related to the mangrove forest structure because there was no significant difference
in the aboveground biomass. Since there was no difference in the species composition,
it is expected that the belowground biomass would not be different either (Table 4).
Soil moisture in the Mojuim River mangrove forest negatively influenced $CO_2$ flux in
both seasons (Table 5). However a correlation with the flux of $CH_4$ was not identified.
Studies show that $CO_2$ flux tends to be lower with high soil saturation (Chanda et al.,
2014; Kristensen et al., 2008). A total of 395 Mg C ha$^{-1}$ was found at the soil surface
(0.15 m) in the mangrove of the Mojuim River estuary, which was slightly higher than
the 340 Mg C ha$^{-1}$ found in mangroves in the Amazon (Kauffman et al., 2018), however
being significantly 1.8 times greater at the low topography (Table 3). The finer soil
texture at the low topography (Table 2) reduces groundwater drainage which facilitates
the accumulation of C in the soil (Schmidt et al., 2011).
4.4   Biogeochemical parameters
Chemical parameters of the soil were better correlated with $CO_2$ efflux in the dry
period, while the C:N ratio, OM, and Eh were correlated with $CO_2$ efflux in both
seasons (Table 5). During the seasonal and annual periods, $CH_4$ efflux was not
correlated significantly with chemical parameters (Table 5), similar to the observed in
another study (Chen et al., 2010). Increased soil moisture reduces gas diffusion rates,
which directly affects the physiological state and microbial activities, by limiting the
supply of the dominant electron acceptors, such as oxygen, and gases such as $CH_4$
(Blagodatsky and Smith, 2012). The importance of soil moisture was evident in the
richness and diversity of bacterial communities in a study comparing the different pore
spaces filled with water (Banerjee et al., 2016). Furthermore, sulfate reduction in
flooded soils (another pathway of organic matter metabolism) is dependent on the redox
potential of the soil. However no sulfate reduction occurs when the redox potential has
values above -150 mv (Connell and Patrick, 1968). In our study Eh was above 36.0 mV,
this indicates that sulfate reduction probably did not influence the OM metabolism.
On the other hand, increasing soil moisture provides the microorganisms with essential
substrates such as ammonium, nitrate, and soluble organic carbon, and increases gas
diffusion rates in the water (Blagodatsky and Smith, 2012). Biologically available



nitrogen often limits marine productivity (Bertics et al., 2010), and thus can affect the
fluxes of $CO_2$ to the atmosphere. A higher concentration of Cmic and Nmic in the dry
period (Table 3), both in the high and low topographies, indicated that microorganisms
are more active when the soil spends more time aerated in the dry period (Table 3), the
period in which the high tides produce anoxia in the mangrove soil. Additionally, the
C/N ratio was well below 40, indicating that soil microorganisms and roots do not
compete for nitrogen (Stevenson and Cole, 1999).
Sulfate-reducing bacteria ($SO_4^{2-}$) are important diazotrophs in coastal ecosystems and
can contribute with significant nitrogen ($N_2$) fixation in mangrove ecosystems (Bertics
et al., 2010; Shiau et al., 2017; Welsh et al., 1996). The negative correlation between
TC, NT, C/N, and MO, along with the positive correlation of Nmic with soil $CO_2$ flux
(Table 5), in the dry period, indicates that microbial activity is a decisive factor for $CO_2$
efflux (Poungparn et al., 2009). The high MO concentration at the two topographic
heights (Table 3), at the two stations studied, and the respective negative correlation
with $CO_2$ flux (Table 5) confirm the importance of microbial activity in mangrove soil
(Gao et al., 2020). Also, $CH_4$ produced in flooded soils can be converted mainly to $CO_2$
by the anaerobic oxidation of $CH_4$ (Boetius et al., 2000; Milucka et al., 2015) which
may contribute to the higher $CO_2$ efflux in the Mojuim River estuary compared to other
tropical mangroves (Rosentreter et al., 2018c). The belowground C stock is considered
the largest C reservoir in mangrove ecosystem resulting from the low rate of OM
decomposition due to flooding (Marchand, 2017).





Table 5. Correlation coefficient (Pearson) of $CO_2$ and $CH_4$ fluxes with chemical parameters of the soil in a mangrove area in the Mojuim River
estuary

| Gas Flux ($g\ m^{-2}\ d^{-1}$) | Season | $C_T$ ($g\ kg^{-1}$) | $N_T$ ($g\ kg^{-1}$) | $C_{mic}$ ($mg\ kg^{-1}$) | $N_{mic}$ ($mg\ kg^{-1}$) | C/N | OM ($g\ kg^{-1}$) | Sal (ppt) | Eh (mV) | pH | Moisture (%) |
|---|---|---|---|---|---|---|---|---|---|---|---|
| $CO_2$ | Dry | -0.68** | -0.59* | 0.18NS | 0.61* | -0.66** | -0.67** | -0.07NS | 0.51* | 0.21NS | -0.49* |
| | Rainy | -0.44NS | -0.20NS | -0.15NS | -0.32NS | -0.50* | -0.63** | -0.54* | 0.53* | 0.47NS | -0.54* |
| | Annual | -0.50** | -0.35* | -0.18NS | 0.00NS | -0.53** | -0.48** | -0.30NS | 0.39* | 0.23NS | -0.56** |
| $CH_4$ | Dry | 0.30NS | 0.07NS | -0.14NS | -0.24NS | 0.34NS | 0.02NS | -0.04NS | -0.38NS | 0.26NS | 0.26NS |
| | Rainy | 0.05NS | -0.09NS | 0.44NS | -0.27NS | 0.09NS | -0.11NS | -0.04NS | -0.13NS | -0.07NS | 0.04NS |
| | Annual | 0.04NS | -0.10NS | -0.01NS | -0.18NS | 0.08NS | -0.01NS | -0.17NS | -0.21NS | -0.08NS | 0.02NS |

NS= not significant; * significant effects at $p \le 0.05$; ** significant effects at $p \le 0.01$



The higher water salinity in the dry season (Table 2) seems to result in a lower $CH_4$ flux
at the low topography, more influenced by the tidal movement in this season (Dutta et
al., 2013; Lekphet et al., 2005; Shiau and Chiu, 2020). Another essential factor for the
reduced $CH_4$ emissions is when $SO_4^{2-}$ in the brine affects the competition between $SO_4^{2-}$
reduction and methanogenic fermentation, because the sulfate-reducing bacteria are
more efficient in hydrogen utilization than the methanotrophic bacteria (Abram and
Nedwell, 1978; Kristjansson et al., 1982). At high $SO_4^{2-}$ concentrations methanotrophic
bacteria use $CH_4$ as an energy source and oxidize it to $CO_2$ (Coyne, 1999; Segarra et al.,
2015), increasing the $CO_2$ efflux and reducing the $CH_4$ efflux (Megonigal and
Schlesinger, 2002; Roslev and King, 1996). This may explain the high $CO_2$ efflux found
throughout the year at the high and, especially, at the low topography (Figure 4). Only
in the rainy season was a significant correlation recorded between salinity and $CO_2$ flux.
Still, in all seasonal periods the correlation between salinity and $CO_2$ and $CH_4$ fluxes
were negative.

Studies in other coastal ecosystems have recorded that methanotrophic bacteria can be
sensitive to soil pH, and reported an optimal growth at pH ranging from 6.5 to 7.5
(Shiau et al., 2018). The higher soil acidity in the Mojuim River wetland (Table 2) may
be inhibiting the activity of methanogenic bacteria by increasing the population of
methanotrophic bacteria, which are efficient in consuming $CH_4$ (Chen et al., 2010;
Hegde et al., 2003; Shiau and Chiu, 2020). In addition, the pneumatophores present in
*R. mangle* increase soil aeration and reduce $CH_4$ emissions (Allen et al., 2011; He et al.,
2019). Spatial differences (topography) in $CH_4$ emissions in the soil can be attributed to
substrate heterogeneity, salinity, and the abundance of methanogenic and
methanotrophic bacteria (Gao et al., 2020). The high Eh values found in both
topographies, mainly in the dry period (Table 2), are unfavorable for $CH_4$ emission. Soil
Eh above -150 mV was considered limiting for $CH_4$ production (Yang and Chang,
1998). Increases in $CH_4$ efflux with reduced salinity were found due to intense
oxidation or reduced competition from the more energetically efficient $SO_4^{2-}$ and $NO^{3-}$
reducing bacteria than the methanogenic bacteria (Biswas et al., 2007). This fact can be
observed in the $CH_4$ efflux in the mangrove of the Mojuim River, because in the rainy
season (Figure 4), when there is a reduced water salinity (Table 2) due to increased
precipitation, there was an increased $CH_4$ production, especially in the low topography



(Figure 4). However, we did not find a correlation between $CH_4$ efflux and salinity, as
already reported (Purvaja and Ramesh, 2001)
No significant correlations were found between $CH_4$ efflux and the chemical properties
of the soil in the mangrove of the Mojuim River estuary (Table 5). However, with an
average flux of 4.70 mg C $m^{-2}$ $h^{-1}$ and with extreme monthly and seasonal variation,
more detailed studies are needed on $CH_4$ efflux and on the relationship with
methanotrophic bacteria and interactions with abiotic factors (mainly ammonia and
sulfate).
**5    Conclusions**
Between latitude 0° to 23.5° S the most recent estimate shows an emission of 2.3 g $CO_2$
$m^{-2}$ $d^{-1}$ (Rosentreter et al., 2018c). However, the efflux in the mangrove of the Mojuim
River estuary was 6.7 g $CO_2$ $m^{-2}$ $d^{-1}$. For the same latitudinal range, the authors
estimated an emission of 0.64 g $CH_4$ $m^{-2}$ $d^{-1}$, and we found an efflux of 0.13 g $CH_4$ $m^{-2}$
$d^{-1}$. Seasonality was important for $CH_4$ efflux but did not influence $CO_2$ efflux. Due to
the rainfall variation compared to the climatology, the differences in fluxes may be an
effect of global climate changes on the terrestrial biogeochemistry at the plant-soil-
atmosphere interface, making it necessary to extend this study for more years. Using the
factor of 23 to convert the global warming potential of $CH_4$ to $CO_2$ (IPCC, 2001), the
$CO_2$ equivalent emission was 35.4 Mg $CO_{2-eq}$ $ha^{-1}$ $yr^{-1}$.
Microtopography should be considered when determining the efflux of $CO_2$ and $CH_4$ in
mangrove forest in the Amazon estuary. The low topography in the mangrove forest of
Rio Mojuim contained a higher concentration of organic carbon in the soil. However, it
did not produce a higher $CO_2$ efflux because this was negatively influenced by soil
moisture, which was indifferent to $CH_4$ efflux. MO, C/N ratio, and Eh were critical in
soil microbial activity, which resulted in a variation in $CO_2$ flux during the year and
seasonal periods. In this sense, physicochemical properties of the soil are important for
$CO_2$ flux, especially in the rainy season; however, they did not influence $CH_4$ fluxes.
*Data availability*: The data used in this article belong to the doctoral thesis of Saul
Castellón, within the Postgraduate Program in Environmental Sciences, at the Federal
University of Pará. Access to the data can be requested from Dr. Castellón
(saulmarz22@gmail.com), which holds the set of all data used in this paper.





*Author contributions:* SEMC and JHC designed the study and wrote the article with the help of JFB, MR, MLR, and CN. JFB assisted in the field experiment. MR provided logistical support in field activities.

*Competing interests*: The authors declare that they have no conflict of interest

*Acknowledgements*: The authors are grateful to the Program of Alliances for Education and Training of the Organization of the American States and to Coimbra Group of Brazilian Universities, for the financial support, as well as to Paulo Sarmento for the assistance at laboratory analysis, and to Maridalva Ribeiro and Lucivaldo da Silva for the fieldwork assistance. Furthermore, the authors would like to thank the Laboratory of Biogeochemical Cycles (Geosciences Institute, Federal University of Pará) for the equipment provided for this research.

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
