# Peer review of "Greenhouse gas fluxes in mangrove forest soil in an Amazon estuary"

_Biogeosciences, 2021_

## Author Comment (AC1)

Main Points

The manuscript presents a study investigating monthly $CO_2$ and $CH_4$ fluxes from mangrove soils in the Amazon across contrasting seasons (wet and dry) and topographies (high and low). This study is suitable for publication in Biogeosciences and is relevant to the community, especially with the focus on spatial and seasonal fluxes. The methods used are robust and the data presented support the conclusions. However, the paper structure and the presentation of the data could be improved.

We are very grateful for your corrections and observations. We think your considerations and, especially, corrections are very important. We hope they have met your expectations, but if you still need more effort from us, please do not hesitate to contact us.

The manuscript suffers from lack of structure, presentation of results in the discussion but not the results section (e.g. correlations), repetition of results in the discussion, and a lack of synthesis of previous literature findings with current study findings. This makes the discussion hard to follow and understand. Please rewrite the discussion particularly addressing the issue of stating facts from the literature without giving context on how they are relevant or improve understanding of your findings, removing results into the results section, deleting those sections which are repetitive of the results, and rearranging to include all GHG discussion first before moving onto the C stored in biomass. Additionally, the discussion would really benefit from discussing results in context of explaining observations, drivers through correlations and comparisons to other studies in terms of magnitude of fluxes but also patterns or drivers found.

We use correlations to try to explain the fluxes, so Table 5 was included in the discussion and not in the results. We also changed the position of carbon biomass discussion in the text. We understand that each result or significant correlation observed in this study was compared with results found in the literature. However, we preferred the results found only in the tropics.

Additionally, the authors should go through the manuscript and ensure there are no grammatical issues or places where too many words are used that make the sentences difficult to understand. With this, the manuscript would be much easier to understand for the reader.

We try to improve as much as possible, and for that, we hire a sworn translator.

Scientific Points

A lot of space is given to the aboveground biomass and carbon stored there, however, this is not the main point of the paper and provides minimal impact. It would be better to tie this into the offset from the GHG emissions in the discussion to justify the value in the paper. For example, calculating how much carbon is stored and how much is emitted through soil emissions, however, to do this the best way you need belowground C stocks too and I don't think you measured these?

Thank you very much for the comment and you are right that we do not measure carbon below ground. Below ground biomass was cited in methods (line 165) to say that we

run away from aerial roots when we install the flow chambers; in line 285 to show how the biomass of Macaca Island is different from other mangrove forests (including Table 4, line 286), and in the discussion to show that the differences in fluxes are not caused by differences in tree structure between the two areas, as these are equal. We also add that information on carbon biomass in aboveground biomass serves as an additional source for comparison with other works carried out outside the Amazon region.

It is not clear how the statistical analysis was performed as it is not clear how the data was averaged. For the seasonal data is it average monthly fluxes in the wet compared with average monthly fluxes in the dry season? For the spatial comparison it is stated in Line 182 'between the different sites' but is this the sampling sites or between high and low topographies? It is also not clear when a t-test and when an ANOVA was used, the caption for Table 1 only mentions Kruskal-Wallis. If a t-test was used then this should be paired when comparing seasons because it is the same sampling sites being investigated. A Pearson correlation was used to determine relationships between gas fluxes and soil properties, however, gas fluxes were taken monthly and soil properties only once per season. Were the correlations performed on seasonally averaged gas flux data?

Thank you very much for your questions, as they were important to improve the text. I will answer your questions below:

**For seasonal data, are monthly average wet season flows compared to average monthly flows in the dry season?**

Yes, monthly data for each station has been grouped together for comparison. In Table 1 we show the monthly fluxes, comparing topographies for each month (n = 16), and different lowercase letters show significant statistical variation. Also, in each seasonal period, we compare the topographies (n = 96), and statistical comparisons are also made with lowercase letters. Capital letters compare seasonal flow (n = 96) within each topography.

**For spatial comparison it is indicated in Line 182 'between different locations' but are these the sampling locations or between high and low topographies?**

Thanks for the question. We improved the text.

**It is also unclear when a t test and when an ANOVA was used, the legend of Table 1 only mentions Kruskal-Wallis. If a t-test was used, it must be paired when comparing stations, because it is the same sampling sites that are being investigated.**

We used the t-test to verify if there was variation between treatments where the distribution of results was normal. When we found a significant variation between the means, we used ANOVA with Tukey's test analysis, according to the statistical program used by us. Only the gas flows did not have a normal distribution, the other analyzes presented in the work showed a normal distribution.

**A Pearson correlation was used to determine the relationships between gas flows and soil properties, however, gas flows were measured monthly and soil properties only once per season. Have correlations been performed on seasonally averaged gas flow data?**

Thank you for asking, we have improved this in the text and hope it is clearer. Correlations were made in the months in which the soil samples were collected, both in the dry season and in the rainy season.

Data from Table 1 would be better presented in a graph instead. I think this is Figure 4 but why is that so far through the paper? The point of the paper is looking at CO2 and CH4 fluxes and I cannot see them clearly presented until the discussion. I think it is really great to provide actual flux values as lots of studies do not, but these could be presented in the SI.

We thought it best to present a table with the mean values and standard error of the mean in the results to facilitate comparison with other papers. However, in the discussion, we chose to make a graph because visually it is easier to compare with the tide. I don't know what SI means.

I suggest presenting CH4 fluxes as mg m-2 d-1 as the values are very small as g m-2 d-1

We chose g m-2 d-1 to give a visual comparison with CO2 fluxes, and we also found some articles published in this same way.

Technical Points

Line 16 – contrasting topographical gradient should be replaced with contrasting topographies

We accept the suggestion and change the text.

Line 18-20 – this sentence is confusing and has some grammar issues, please rewrite. This is often true through the manuscript so please read through and improve sentence structure throughout to aid understanding.

We accept the suggestion and change the text.

The abstract does not include the fluxes of CO2 and CH4, instead the variation between topographies and season is first discussed. Please include the values first and then state the variation.

We accept the suggestion and change the text.

Line 21-22 – These mention CH4 fluxes between high and low topography but are contradictory. Please rewrite with the correct observations.

We accept the suggestion and change the text.

Line 24-26 – This sentence is confusing and needs rewriting with more context, you are stating the GWP of the mangroves through CO2-equivalents but it is not clear.

We accept the suggestion and change the text.

Line 29 – replace macro tide with macrotidal

We accept the suggestion and change the text.

Line 38-40 – This does not fit with the previous sentence and perhaps needs moving earlier in the paragraph

We accept the suggestion and change the text.

Line 51 – The flux unit needs spaces between g, C and yr

We accept the suggestion and change the text.

Line 52-66 – This could be restructured so that the estimates of CO2 and CH4 emissions from tropical mangroves are presented together and then the effects of flooding as a control on CO2 and CH4 fluxes discussed.

Related to the point above, there are some issues with the structure of the introduction making it difficult to follow and preventing the knowledge gaps and importance etc being clearly presented.

We accept the suggestion and change the text.

Line 60-61 – Sulphate reduction generally inhibits methanogenesis, as you discuss in the discussion, but here you say sulphate reduction increases CH4 formation.

We take this information from Purvaja et al. (2004), who wrote the following sentence: ...Methanogenesis is the prevalent terminal process in anoxic freshwater sediments and sulfate reduction in anoxic marine sediments.

Line 67-71 – I think the objectives of the study could be clearer. Spatial variation is being investigated but it is specifically high versus low topography, I would add an extra sentence stating this and that the seasonal variation captures wet and dry seasons. This can then lead to flooding and salinity being key controls you are looking at. Also mention the controls of soil properties on these fluxes are being investigated.

We accept the suggestion and change the text.

Line 68 – pristine may be a better description than non-anthropized

We accept the suggestion and change the text.

Line 77 – coastal strip, should this be a coastal strip/area of mangroves?

This was done; mangrove area

Line 78 – Am type, best to say tropical monsoon

We thought it best to continue with the classification used to define the tropical monsoon climate, as it is better known in climatology, and we put in parentheses the classification of tropical monsoon climate.

Line 90-91 – Average mean salinity – is that in the river water or in the mangrove sites?

As can be seen in Figure 1, the Mojuim River forms the estuary of the São Caetano de Odivelas region. In the month where the greatest amounts of rain occur, the water is less salty, and in the months where the rains are less, the river water is saltier. Throughout the year the waters of the places where the measurements were carried out (mangrove) were salty.

Line 106 – Four sampling sites - are these what you later call stations? It was not clear to me in the paper if the sampling sites were later referred to as stations or if you are referring to something else? When you state in the results there was no difference between station are you comparing low and high topography or sampling sites at the same topography?

We changed the term sampling site to plot and we think that the sampling design may have become clearer.

Line 110 – flux for each chamber was measured – at this point it is not clear if a chamber is equivalent to a sampling site, or if there are multiple chambers at each sampling site. I think you can just say gas fluxes were measured during periods…

We changed the term sampling site to plot.

Line 111 – states when the low topography was measured, was the high topography measured at the same time or when was this measured?

This was done

Line 114 – replace 'by a macro tide dynamics' with 'by macrotidal dynamics'

This was done

Line 119 – Describe the sampling locations before this section, and then have these subsections talking about what was measured and how. You reference here being the same as the gas flux sites, but we haven't had details of those yet so I think presenting the sampling locations/plots first will make everything clearer

This was done

Line 129 – Soil sampling and environmental characterisation was measured once during wet and dry season, so when the Pearson correlations were calculated, was this done with seasonal flux data? I don't think this was very clear so please state this in the statistical analysis section.

Line 133 – soil samples were properly stored – please give details instead of writing this.

Line 152 – You switched from gas flux measurement to flow measurement, be consistent throughout and use gas flux measurement.

Line 152 and 155 – Suggest using $T_{air}$ with air as subscript, same for $T_s$ for soil temp

Line 156 – Remove sequentially

Line 159 – Should be flux measurements

Line 161 – how were plots randomly selected?

The plots were selected according to the topography, that is, in the low topography, where the tide entered every time, and in the high topography, where it almost did not enter. The random selection was for the locations of the flux rings, that is, in a circumference of 10 m radius we threw the PVC rings and where they fell they were placed.

Line 169 – Please give details of the standard gas used for calibration

Line 177-178 – So the 6% of data with weak regression were considered as zero?

Line 182 – different sites, is this between the high and low topography?

This was done: Yes, between topography

Line 186 – Remove 'and' after with.

Line 188 – Please add the details of the relationships between gas fluxes and soil properties to the results section.

We did this in the discussion of results, especially between lines 420-428.

Figure 2 – Axis label Dez should be Dec

This was done

Line 200 – Rearrange the gas flux data to first present the mean and range of the fluxes and then discuss the stats and differences spatially and seasonally.

In the description of the flux results, which are in table 1, we only emphasize the differences that were statistically significant, without bothering about presenting the mean and the variation, as these values are in the table. The discussion of the results is in the item below.

Line 203 – Should this say only differed significantly?

This was done

Line 203 – Replace among with between

This was done

Table 1 – The presentation of the stats in this table is very confusing. Do lowercase letters compare monthly fluxes between high and low topography, and uppercase letters compare dry versus wet seasonal data for each topography? As stated above I do not know what the stations refer to and so I don't understand some of the comparisons – maybe the uppercase letters are not comparing dry versus wet but something to do with the stations?

Lowercase letters compare topographies in the same month, that is, compare the lines for each gas separately. The capital letters compare the stations in each topography, that is, it only compares the two seasons for the High and Low topography, separately.

Line 213 – Add here that this is seasonal data comparison now

Paragraph on line 222 – This would be clearer if it was rearranged to start saying greenhouse gas fluxes were only sig different between topograpohies in the dry season where co2 fluxes were higher at the high topography and ch4 fluxes were higher at the low topography.

This was done

Line 225 – I am not sure what you refer to here with 'with this' and so I am not sure if you are presenting here the fluxes over the dry season or across all topographies?

We made it clearer: In the high topography, the mean annual fluxes ….

Line 229-242 – Here you sometimes say high versus low topography and sometimes between stations. As mentioned earlier I am not clear on what the stations are referring

to, in any case it would be best to stick with the same naming e.g. always talking about high versus low topography.

This was done: We changed station to seasonal, and made it clearer in the text where you pointed out the difficulty of understanding.

Line 234 – remove variable

This was done

Line 249 – Replace CT with TC, here and elsewhere

This was done

Line 258 – Replace MO with OM, here and elsewhere

This was done

Line 266 – Tar should be Tair?

This was done

Line 271 – Vv is not defined, please define here

Thanks for the observation, the translation was not correct, as Vv means wind speed (Ws). This has been modified in the text.

Line 292 – Replace e with and

This was done

Line 293-294 – I think you mean to say rainier than long-term average in the dry and less rainy in the wet season than the long-term average, because on figure 2 ppt is higher in the wet and lower in the dry.

This was done: It is important to consider that when compared with the climatological average (1981-2010), the year under study was rainier in the dry season (2017) and less rainy in the wet season (2018).

Line 298 – Here you present the total carbon rate – do you mean the total carbon fluxes? The units are in CO2 not C so I am not sure how this is carbon flux. How was this total emission calculated? Through converting CH4 emissions to CO2-equivalent emissions using sustained global warming potentials?

We calculated the weight percentage of C in each molecule (CO2 and CH4) and multiplied by the fluxes separately (1.82 g C/m2/d for CO2; and 0.10 g C/m2/d for CH4) and made the sum of two values. We corrected it in the text because when the translation was performed there was an error in the values.

If figure 4 is the data from table 1 plotted onto graphs, then why are there no statistical differences presented on here?

In Table 1 there was an error in the statistical analysis, as the CO2 fluxes were not different in the dry season, however the fluxes were different in the rainy season. We redone the statistical analysis, and the other results are correct.

Figure 4 – I suggest a different naming scheme for topography, using T_high and T_low I am automatically thinking of temperature and then I was trying to work out what the temperature differences were.

This was done

Line 315-316 – Why not compare the annual CH4 flux, like you did for CO2?

This was done

Line 314-323 – Here it is stated that CO2 is higher than literature values and CH4 is lower but this is just stated with no discussion of why this may be true, for example, do you have less flooding due to combining the low and high topographies in this analysis? Is it due to soil properties here being less favourable for methanogenesis and more favourable for aerobic respiration?

This was discussed below.

Line 324 – Here the discussion on biomass is in the middle of all the gas flux discussion, move this to after the gas flux has been discussed.

This was done

Line 333-334 – I'm not sure how your results show the mangroves are more productive than previously known. You have C storage at lower capacity than estimated for brazilian mangroves, then state a primary production for tropical mangrove forests with no relation to your measurements here (unless I missed it) and then say the mangroves are more productive than previously known.

This was done: The sentence that addressed mangrove productivity was removed, as we did not study this, and we added the following sentence: The biomass found in the Mojuim River estuary does not seem to be different from the biomass found in other Amazonian mangroves, however much smaller than that found in other mangroves Brazilians.

Line 372-374 - This sentence is very hard to follow. This is an example of where the whole manuscript could benefit from another proofread to check for clarity. Additionally, to reduce sentence length where unnecessary text is used that makes the point of the sentence harder to understand as the reader.

This was done: The results show that the physical parameters do not act in the fluxes in a standardized way, and their influence depends on the topography and seasonality.

Line 379 – High tide or rainy season? Because all fluxes are measured at low tide I think. If this is not correct please make this clear.

This was done: …especially during the rainy season when the tides were higher.

Line 388 – replace generates with favours

This was done

Line 389-392 – Please also rewrite this sentence

It was rewritten.

Line 409-410 – Better correlate dto which characteristics?

Thank you for the question. We rewrite the sentence: The dry season was the period in which we found the greatest amount of significant correlations between CO2 efflux and soil chemical parameters.

Line 410 – Positive or negative correlations?

Here we are reporting that only these soil chemical variables (C/N ratio, OM, and Eh) correlated with CO2 flux in both seasons. However, when we look at Table 5, we see that in both seasons the correlation of the C/N ratio and OM with CO2 was negative, and the correlation of Eh with CO2 was positive.

Line 415 - Here this is an interesting point but it seems you are saying that higher soil moisture should give a lower ch4 efflux, but you also show and state earlier that lower topography with more flooding has higher ch4 due to anoxic conditions. I think it would be good to really refine these points and discuss them together. This is another example of where there is lots of comparison and citations of other literature but it is not always pulled together in a coherent way.

As can be seen below, and in the text, it has already been corrected.

Line 424-425 – Here you say increasing soil moisture increases gas diffusion rates but earlier you said high soil moisture decreases gas diffusion rates

We accept. Line 412 - The soaking of the soil reduces gas diffusion rates …

Line 430 – DO you mean here that during the dry season the high tides cause anoxic soil conditions, or are you comparing dry and wet seasons?

Thank you for the question. We are saying that it is the period when only high tides produce anoxia in mangrove soil with low topography. We improved the text.

Line 430-432 – I am not sure what this is relevant to.

We accept.

Line 433 – I am not sure why this is relevant, you are not discussing sulphate reduction but CO2 and CH4 fluxes, and this sentence does not seem to link to your next points clearly.

We accept.

Line 441-444 – Is there a reason that this same mechanism would not be occurring in the mangroves you are comparing your results with?

Thank you for the question, but I'm not saying that in other places this mechanism is not happening, I'm just saying that in the area we studied this mechanism may be more intense.

Table 5 gives correlation coefficients with annual data, so it would be good to plot the annual fluxes onto figure 4.

We accept.

Line 451 – This was not clear to me. Are you saying that tidal movement is more important for flooding in the dry season, therefore, there is also higher salinity?

No, we are just saying that due to less precipitation in the dry period, there is less influence of the river in the estuary, consequently, the waters become saltier. Likewise, as it has a smaller volume of water, only the low topographies are periodically visited by the tides.

Line 462-463 - Here you say salinity is negatively correlated with CO2 but this paragraph earlier states that high sulphate leads to increased CO2. There are lots of these instances and I think the discussion could really benefit from more structure and focus into the synthesis of previous work on drivers and patterns related to this study, rather than stating lots of observations from the literature and then saying in this study we found X.

Your observation was very important, however, our results showed that there was only a significant negative correlation with CO2 in the rainy season (Table 5), and in this sense; we decided to remove the sentence that is bringing doubt. We think that is important to leave the discussion on sulfate in the text because with this information we have one more source of CO2 for the atmosphere, and not just the fluxes being the result of a metabolic function of respiration, and decomposition of organic matter.

Line 498-500 – Consider using the sustained global warming potential instead – moving beyond global warming potentials to quantify the climatic role of ecosystems. Scott Neubauer and J. Patrick Megonigal, Ecosystems. 2015

We accepted and it was added to the text.

---

## Author Comment (AC2)

**General comment**

The manuscript "Greenhouse gas fluxes in mangrove forest soil in the Amazon estuary" examines $CO_2$ and $CH_4$ fluxes from mangrove soils and evaluates topographic and seasonal variations. Moreover, environmental drivers such as vegetation structure and soil characteristics are studied. I highly agree with the comments made by RC1. The topic of the paper is timely and fits the scope of Biogeosciences. The study design is appropriate, and the data set sufficient to answer the stated research questions. However, the manuscript is hard to follow and should be streamlined. It would be helpful to put some of the results (e.g., detailed statistical analysis of all parameters) into a supporting information and report only relevant findings. The general structure and story line should be more focused.

We think your considerations and, especially, corrections are very important. We hope they have met your expectations, but if you still need more effort from us, please do not hesitate to contact us.

**Specific comments**

Abstract

L 13 – 14 First time reading "especially in a scenario of global climate change" I thought you would test the impact of climate change on GHG in mangroves. Maybe rephrase as "to asses their impact on climate change".

In the abstract, the inclusion of the phrase *in a global change scenario* was suggested by an editor, as the article will be included in this topic. In an attempt to improve the understanding of the phrase, we changed it to: particularly in the current scenario of global climate change.

L 17 – 18 Delete this part of the sentence and use the extra words to give more quantitative results, such as gas fluxes.

We change for: The plots for the trace gases study were allocated at contrasting topographic heights.

L19 – 22 Write how much higher (x-times higher) fluxes were between sites/seasons. Do never start a sentence with "only". Chance this throughout the manuscript.

We rewrote the phrase and included the found values: In the spatial variation, the $CO_2$ flux was greater in the high topography (7.858 g $CO_2$ m-2 d-1) compared to the low topography (4.734 g $CO_2$ m-2 d-1) in the rainy season, and the $CH_4$ flux was greater in the low topography (0.128 g $CH_4$ m-2 d-1) than in the high topography (0.014 g $CH_4$ m-2 d-1) in the dry season.

Introduction

L28 – 30 Move this sentence to the study site description or the aim paragraph of the introduction. Then start with (tropical) mangroves in general. Consider restructuring the first paragraph by starting with carbon storage in mangroves and benefits for climate

change. Then state that it is important to consider GHG outgassing as offset of the carbon storage.

This has been done.

L41 Change "attributable" to "driven by".

This has been done.

L50 Write "CO2 outgassing" instead of "CO2 production to the atmosphere". Make clear which statements in this and the next paragraph are specific to mangrove and which to estuaries/coasts/vegetated coastal wetlands. Preferably use only mangrove publications, there are enough publications to underline each of your statements.

This has been done. Alongi's book(2009) presents a functional overview of mangrove forest ecosystems; how they live and grow at the edge of tropical seas, how they play a critical role along most of the world's tropical coasts, and how their future might look in a world affected by climate change. The study of Rosentreter et al. (2018a) quantifies seasonal pCO2 and CH4 concentrations and emissions along the salinity gradient of three tropical mangrove-dominated estuaries in Australia.

L55 Confusing statement. Consider rephrasing.

This has been done.

L61 How does reduction of sulfate produce CH4?

Purvaja et al. (2004) wrote: One expects a stimulation of methanogenesis and hence of methane emission during monsoon, because the impact of freshwater should shift the electron flow from sulphate-reducing bacteria to methanogens. We have changed the text and hope it is clearer now

L67 Be more specific what you mean by spatial and seasonal variation.

This has been done.

L70 Remove years but describe in more detail which drivers you were testing.

This has been done. We don't quite understand what you mean by "drivers"

Methods

Please add GPS coordinates of your stations in the text.

This has been done.

L77 "exclusively untouched mangrove forests" use "pristine mangroves". Consider splitting this sentence.

This has been done.

L86 Use tidal "amplitude" instead of "height".

This has been done.

L109 – 111 This sentence is not a part of the study site. Put it into "Flux measurements". Is suggest putting "Greenhouse gas flux measurements" as 2.2, since this is your focus.

This has been done.

L120 When did you conduct the floristic survey? Report dates.

This has been done.

L130 Why did you take only very shallow soil cores? It sounds like you measured pH and redox at the same spot where you took the soil sample. I hope it was just next to it. Please clarify.

It is clear that the pH and redox potential measurements were performed prior to collecting soil samples for the laboratory, ie with intact soil. We used a depth of 0.10 m because from the literature review this is where the fluxes happen.

L137 I personally would not capitalize all parameters, but wright "Organic matter…".

We think it is important to specify and not generalize as being just MO, as these parameters will appear in the results.

I agree with RC1 that the abbreviations should be changed.

This has been done.

L150 When did you conduct the soil sampling? Report dates.

This has been done.

L160 Also add dates of chamber sampling.

Sorry, I don't understand, do you want to put all the dates that the flows were measured? Flows were measured every month from July 2017 to June 2018, on waning or waxing moons.

L164 I personally would have measured above the mangrove roots since these are important parts of the mangrove ecosystem. At healthy mangroves, spots without roots are rare, thus including them yields more representative flux rates for mangrove soils. Something to consider in your next study.

Thanks for the observation, we will take it into account for the next study.

You need to add more info about the flux measurements. How often did you measure per month? One or more rings? Did these rings stay at the same spots?

Flows for each chamber (total 8 rings per plot) were measured once a month, during waning or waxing moon periods. I added in the text that the rings remained in place until the study was completed.

Consider matching headings with the results headings. The wording and also the order.

This has been done.

Results

The results section is very hard to read. I would only report values of each parameter and describe general trends without using any statistics. Then add a section where you look at the statistics, but only in regard to the GHG not of the statistics between the drivers.

We think statistics are part of the results and important to show where the differences are.

3.1 Carbon dioxide and methane fluxes

- Fig 4 (put table 1 in SI)
- We don't understand what you mean here and what SI means
- Describe values and trends for CO2 and CH4 in sperate paragraphs

We put the flows of CO2 and CH4 in the same paragraph because we think it is important to compare not only how the flux of each gas behaves, but also to compare between them. If we put each one in different paragraphs, the text can get tiring, and difficult to compare the behavior of the two gases.

3.2 Weather data

We think that showing that there is a seasonality in the climate before showing the gas results is important, as this justifies dividing the year into two seasonal periods.

We think that figure 2 and figure 3 show different things, as figure 2 shows the sum of rainfall for each month, during the years from 2017 to 2018 and for the period from 1981 to 2010. Figure 3 shows the measurements carried out at the same time the flows were measured. Placing the two figures in the same place would perhaps be difficult for the reader to understand these differences.

- Fig 2 + Fig 3

3.3 Soil characteristics

If your intention is to join the two tables, it would be very difficult to put on the same page with the richness of details, which we think are necessary for the presentation of the results, and then for the discussion.

- Table 2 + Table 3

3.4 Vegetation structure and biomass

- Table 4

3.5 Drivers of greenhouse gas fluxes

- Table 5 (also add correlation of all other parameters, to shorten the table you could only keep significant correlations and mention in the text which parameters were not significantly correlated to the GHG)
- I would not distinguish between single months for the correlations, but focus only on wet and dry seasons

Alternatively, you could only use subsections 3.1 – 3.4 from above. Start each subsection with describing values and trends of each parameter. The second part of each subsection should briefly report the stats between the GHG and the parameters (not amongst parameters!).

In the same way that we wrote for evaluator 1, the correlation table was used to discuss and try to explain which environmental factors are correlated with the fluxes, and in this sense, we would like to leave table 5 in the discussion.

In all tables use mean ± standard error instead of mean(standard error).

 This has been done.

Discussion

Generally, try to link your results with the literature more closely. Often you have one sentence about one study and then and an vaguely related sentence about your study. You need to link those "bigger/smaller than, similar to, supported by/contradicting to…"

Possibly use the following structure:

4.1 Carbon dioxide and methane fluxes

- Compare fluxes to literature and discuss differences
- Separate CO2 and CH4 in paragraphs

  We put the flows of CO2 and CH4 in the same paragraph because we think it is important to compare not only how the flux of each gas behaves, but also to compare between them. If we put each one in different paragraphs, the text can get tiring, and difficult to compare the behavior of the two gases. We ask if possible to leave it the way it is.

4.2 Drivers of greenhouse gas fluxes

- Possibly get subheadings for parameters similar to results section
- Discuss drivers and find literature backing up your statements, only focus on significant differences but do not repeat statistics

> Statistical correlations between lines 365-372 are not repeated, as they are not shown in table 5.

L 309 Add a, b, c, and d to the figure. Reduce scale for c and d.

This has been done.

L294 – 295 To speculative – delete.

This sentence was requested by an editor, for the article to have a link with the topic where it will be published (global climate changes).

L 295 – 299 Also needed in results section, here only short repetition. What do you mean by total carbon rate? Separate $CO_2$ and $CH_4$.

L 301 Slufate reduction? Explain.

Make sure that all studies you compare your results to used similar methods and did note examine water – air fluxes instead of soil – air fluxes.

Fluctuations in redox potential altered the availability of the terminal electron acceptor and donor and the forces of recovery of their concentrations in the soil, such that a disproportionate release of $CO_2$ can result from alternative anaerobic degradation processes such as sulfate and iron reduction (Chowdhury et al., 2018).

Calculate all GHG fluxes in the discussion in the same unit to make comparisons easier.

The units were placed differently in the text, to facilitate comparison with the cited article. For example: in line 302-304 Shiau and Chiu (2020) published their results in g $CO_2$ m-2 d-1, but Alongi (2009, LN 305-306) published their results in mmol $CO_2$ m-2 h-1. For this reason, we put the units in a different format in the text.

L318 What was expected?

In lines 314-315 it is written that Rosentreter et al. (2018b) estimated production of 26.7 mg $CH_4$ m-2 h-1 for mangrove soils in tropical latitudes (0 and 5°), for this reason, we write that our values were lower than expected.

4.2 Mangrove biomass: would only focus on the impact on GHG. This section interrupts the flow of the manuscript.

This has been done.

L336 Larger flood volume during ebb tides? Explain.

Translation problem, already fixed: Mangrove areas are periodically flooded, with a greater volume in syzygy tides, mainly in the rainy season.

L433 – 435 I do not understand your general focus on sulfate reduction. This is an alternative process to methanogenesis. Always focus on $CO_2$ and $CH_4$ production.

This has been taken from the text.

The difference between different topographies can probably be explained by differences in chemical soil characteristics.

You're right.

L493 Which authors?

This has been done: Rosentreter et al. authors (2018c)

Conclusion

495 – 498 Be more specific. What seasonal trends? Rainfall compared to climatology?

This has been done: Due to the rainfall variation compared to the climatology, that is, rainier in the dry season and drier in the rainy season, the differences in fluxes may be an effect of global climate changes on the terrestrial biogeochemistry at the plant-soil-atmosphere interface, making it necessary to extend this study for more years.

Add a sentence about the general relevance of the study.

---

## Author Response (AR1)

Thank you for responding to the comments posted by the two reviewers, who both raised important points to be considered. Both reviewers agreed, that the study is interesting, of relevance and important to consider, but also both indicated that the authors need to clarify some statistical and technical details, and to improve the structure within the manuscript, especially of the discussion. One thing to clarify, the editor did not request from the authors to put specific sentences into the manuscript, but as the article was submitted to a special issue on 'Global change effects on terrestrial biogeochemistry at the plant–soil interface' the authors should demonstrate that their study is relevant for that special issue.

On 12/23/2021 the editor Lucia Fuchslueger wrote: thank you for pointing out the global change aspect in the discussion and conclusion. It would be great to make it clear as well in the abstract already if this is possible. I understood that you were asking to put a sentence in the summary. In response to the editor's recommendation, I made it clear in some places in the text that the possible differences between the flows that occurred on Macaca Island may be the result of climate changes already underway, given that the summer was rainier and the winter drier in compared to climatology.

Both reviewers suggested to add more details and clarify some parts of the statistics (e.g. which data was used to compare averages, and what exactly was compared in statistical tests). Thank you for clarifying some details on statistics and averages already in your reply.

We added more detail to the statistical design and clarified what data were used to compare means and what exactly was compared in statistical tests.

Moreover, both reviewers suggested to move redundant data (e.g. Table 1) into the SI (which is Supplementary Information for clarification). I fully support this, and also suggest to present the data that is now in Figure 4 already earlier to the results (also to avoid presenting redundant data in manuscripts).

We have moved Table 1 to the SI (supplementary information) as well as Figure 4 has moved to the results part.

Indeed, results should be strictly moved from the discussion section and integrated in the results section. Both reviewers made great suggestions how to re-structure both results and discussion section. As indicated, please use the discussion section to put your GHG gas emission results into context with literature, and follow the suggestion made by the reviewers on changing the structure and sequence within the discussion section. As CO2 and CH4 are the results of very different biological processes I would also suggest to at least put the results into different paragraphs (maybe not sections), but as rein the discussion their dynamics and controls also separately in the discussion. Screen for repetitive sections.

We accept the excellent suggestions from the reviewers by better restructuring the results and the discussion. We also accept the change of structure and sequence in the discussion section. In the same way, the $CO_2$ and $CH_4$ results were placed in different paragraphs.

Thank you for taking your precious time to improve the presentation of the article so that it has a better understanding.

---

## Author Response (AR3)

Dear Editor,

I am very grateful for the extremely thorough review of our work, which made it better and more realistic. In this sense, I answer the questions and suggestions made.

in Line 340 begin of the discussion it would be easy to remember the reader that in fact there was a LaNina happening at the time of the experiment, which is I think really interesting (I think you have mentioned a reference earlier in the results, maybe repeat here again).

Thank you for the great suggestion, and this was accomplished by putting the sentence: *Perhaps this variation is related to the effects of La Niña, and the intensification of extreme events is considered as climate change.*

Line 458: similar as observed in another study.

This has been modified in the text.

Line 517: this somehow is difficult to follow, why are more studies needed due to the fluxes of rates given here? the argumentation is not totally clear to me.

For us, this argument seems to behave, but to avoid complications we decided to withdraw it from the text.

Non-public comments to the Author:
I acknowledge the choice of Kruskal Wallis ANOVA if you have non-normal distributed data. nevertheless linear mixed models, where it can be accounted for repeated measures structure, and would then in the end require normal distribution of residuals (using e.g. qqplots) and may be then even be used and may be accurate.

We understand your comment and are very grateful for the indication of linear mixed models. However, this work resulted from the thesis of my student, who returned to Honduras and is engaged in another work activity. On the other way, we think that local variability is a very important factor to be considered, and trying to intensify the statistical analysis may fail to take this spatial variability into account.

On another note, in is not really possible to use Pearson relations if data is not normal distributed, but rather Spearman correlations. so please check if this was the case or not for the data used for the correlations shown here.

In our results, only the distribution of the two gases was nonparametric, all other variables were of parametric distribution. Pearson's correlation determines the degree to which a relationship is linear. Put another way, it determines whether there is a linear component of association between two continuous variables. As such, linearity is not really an assumption of Pearson's correlation. On the other hand, we transformed the data and had relationships similar to those presented in this work.

https://statistics.laerd.com/spss-tutorials/pearsons-product-moment-correlation-using-spss-statistics.php

https://jbds.isdsa.org/public/journals/1/html/v2n1/p8/

Please again, in Table 2, it seems very interesting that twice analytics gave exact the same numbers AND standard errors for dry and rainy season on the granulometry. Please one more double check if there was not anything happening during editing.

We are extremely grateful for the editor's insistence that we review the data in Table 1. Because my student was out of Brazil, I personally went to the EMBRAPA soil laboratory to request the soil texture analysis. With this, I realized that my student used the same results for the different seasonal analyses. I made the correction in the table and now the data are from the two samples, collected in the two seasonal periods.

---

## Author Response (AR4)

Dear Editor, once again we are grateful for the suggestions to improve the text. All suggestions and corrections have been made and we look forward to continuing with the publication process.

Line 214: you mean 'statistically significant higher..'?

I tweaked the sentence to see if it gets better understood.

Similarly, at the low topography, CO2 fluxes were statistically significantly higher (H = 19.912; p = 0.046) in September and February when compared to January and November, and not differing from the other months.

Line 238: you mean 'resulting in a total carbon efflux rate...'?

Yes, thanks for the suggestion: resulting in a total carbon efflux rate of 1.92 g C m-2 d-1 or 7.00 Mg C ha-1 y-1

Table 1: thank you very much for double checking and clarifying!

I thank you for your insistence on asking to review the table data.

Line 342: do you mean here: this variation could have been related by a La Nina anomaly, or caused by more intense and extreme weather phenomena caused by global climate change?

We have changed the sentence so that it can be understood better: Perhaps this variation is related to the La Niña effects (extreme event), taking into account that the intensification and higher frequency of extreme events result from climate change (Gash et al., 2004).

Line 345: use past tense.

We changed to the past: negative values represented gas consumption

Line 370: use past tense (was insignificant or 'very low' would maybe be better )

This value is was very low compared ...

Line 397: is it important for this comparison to include the numbers of studies here? and please clarify: the 28% are lower in dry compared to wet conditions?

It was clarified and I removed the sampling numbers

Line 534: MO or OM?

In English the correct is OM, thank you.